# FMS PINN: Flow-matching sampling for efficient solution of partial differential equations with source singularities

## Abstract

Singularities in the source functions of partial differential equations (PDEs) can pose significant challenges for physics-informed neural networks (PINNs), often leading to numerical instability and necessitating a large number of sampling points thereby increasing the computational time. In this paper, we introduce a novel sampling point selection method to address these challenges. Our approach is based on diffusion models capable of generative sampling from the distribution of PDE residuals. Specifically, we apply the optimal transport coupling flow-matching technique to generate more sampling points in regions where the PDE residuals are higher, enhancing the accuracy and efficiency of the solution. In contrast to existing approaches in the literature, our method avoids explicit modeling of the probability density proportional to residuals, instead using the benefits of flow matching to generate novel and probable samples from more complex distributions, thereby enhancing PINN solutions for problems with singularities. We demonstrate that this method, in certain scenarios, outperforms existing techniques such as normalizing flow-based sampling PINN. Especially, our approach demonstrates effectiveness in improving the solution quality for the linear elasticity equation in the case of material with complex geometry of inclusion. A detailed comparison of the flow matching sampling method with other approaches is also provided.

## 1 Introduction

Physics-Informed Neural Networks (PINNs) are used to solve Partial Differential Equations (PDEs) using neural networks. With the rapid development of computing resources and machine learning algorithms, PINNs have become popular for a wide range of realistic simulations Raissi et al. (2019) . PINNs utilize automatic differentiation mechanisms to encode PDEs into loss functions, incorporating PDE residuals and boundary conditions. PINNs may be preferred over classical numerical solvers due to their easy coding algorithms for both forward and inverse problems, and their ability to handle high-dimensional problems. Despite the widespread success of PINNs in various PDE-related problems they often struggle with complex PDEs, leading to "failure modes" Wang et al. (2022). Specifically, PINN loss function is very non-convex making it challenging to find a global minimum using conventional optimization algorithms for neural network training, such as Adam. Moreover, according to the F-principle Xu (2020), in PINNs the low frequency features of the solution are captured emerge first, while it will take several training epochs to reproduce high frequency features. In this regard, PINNs may be not efficient for solving PDEs with high-frequency solutions, as shown in Chuprov et al. (2023). For simple PDEs (single-scale, single-mode), conventional PINNs can quickly achieve satisfactory solutions Buzaev et al. (2023). However, for more complex PDEs, conventional PINNs often fall short as the low-frequency global solution deviates from the exact solution. To address these issues, recent years have seen the emergence of efficient implementations of the PINNs method. For instance, loss re-weighting methods McClenny & Braga-Neto (2023) and adaptive sampling strategies Gao et al. (2023) have been developed to find a balance between loss and probability distribution on weight or sampling, enhancing the performance of PINNs in complex scenarios.

The goal of this paper is to design a novel method of point sampling to address these challenges. This approach uses the idea of diffusion models which are capable of generative sampling from the distribution of loss residuals. The optimal transport flow-matching technique is applied to generate more sampling points in regions where the PDE residuals have large values, enhancing the accuracy and efficiency of the solution.

## 2 PINN OVERVIEW

Consider a domain $\Omega$ on which we want to solve a partial differential equation $\mathcal{D}u(\boldsymbol{x}) = s(\boldsymbol{x})$, where $s(x)$ is the source function, $\mathcal{D}$ is a differential operator, while $\boldsymbol{x}$ is a $d$-dimensional vector from $\Omega$. The domain is bounded by $\partial\Omega$ on which solution is subject to the following boundary condition: $\mathcal{B}u(\boldsymbol{x}) = g(\boldsymbol{x})$ for $\boldsymbol{x} \in \partial\Omega$, where $\mathcal{B}$ is the boundary operator (e.g. Neumann or Dirichlet conditions). Thus, we have

$$\begin{aligned} \mathcal{D}u(\boldsymbol{x}) = s(\boldsymbol{x}), \quad \forall \boldsymbol{x} \in \Omega \\ \mathcal{B}u(\boldsymbol{x}) = g(\boldsymbol{x}), \quad \forall \boldsymbol{x} \in \partial\Omega. \end{aligned} \tag{1}$$

We consider a neural network $u_\psi(x)$ that approximates the solution of Equation 1, where $\psi$ represents the parameters of the neural network that will be optimized during the training process. The training of the neural network is based on the minimization of the following function:

$$\min_\psi L(\psi) = L_{PDE}(\psi) + L_{BC}(\psi) = ||r(\boldsymbol{x}_i, \psi)||_{2,\Omega} + ||\mathcal{B}u(\boldsymbol{x}) - g(\boldsymbol{x})||_{2,\partial\Omega},$$

where

$$L_{PDE,N}(\psi) = \sum_{x_i \in \mathbb{S}_k} \left( \mathcal{D}u(x_i, \psi) - s(x_i) \right)^2 \tag{2}$$

and

$$L_{BC,N}(\psi) = \sum_{x_i \in \partial\mathbb{S}_k} \left( \mathcal{B}u(x_i, \psi) - g(x_i) \right)^2. \tag{3}$$

The partial derivatives of PINN with respect to the vector $\boldsymbol{x}$ can be computed on the basis of automatic differentiation libraries.

## 3 RELATED WORK

### 3.1 ADAPTIVE SAMPLING AND ITS RELATION TO GENERATIVE LEARNING

The easiest method for selecting points is a mesh grid (regular grid), that is often used in finite difference schemes. However, in Wu et al. (2023) it was shown that this approach can potentially yield trivial solutions and that the PINN solution derived on a uniform grid is more accurate than that obtained on a mesh grid. In addition, sampling based on pseudo-random series (e.g. Sobol sequences, Latin hypercube etc) can be utilized.

Adaptive sampling methods are based on the principle to select points based on their influence on the loss function. One of the algorithms that pioneered this approach is the so called Residual Aided Refinement (RAR) algorithm Lu et al. (2021). The RAR algorithm aims to improve the distribution of residual points during training by introducing additional points in areas where the PDE loss values are large after a certain number of iterations. An advanced version of RAR is called residual adaptive distribution algorithm: RAD Wu et al. (2023). The PINN training begins with uniformly distributed points. After a few iterations, residual values are evaluated, and new points are added in areas with high residuals. The PINN model is retrained with the updated set of points, and this process is repeated to improve accuracy. This algorithm is similar to the classical importance sampling method, an extension of Monte-Carlo methods.

The algorithm that apply importance sampling idea to estimation of loss and sampling points for PINN is Nabian et al. (2021), where they propose to a special proposal distribution that is based on calculation of residual error at nearest to a point specially selected seed points. For instance, the proposal is a distribution that is a PDE residual loss at nearest seed point divided by sum of all

residual values. It means that points with larger values of PDE residual are more likely to be added to a batch. A similar methods are presented in Nabian et al. (2021).

One of the first methods introduced for sampling for variance reduction purposes was normalizing flows. In image generation such methods as GANs Goodfellow et al. (2020), VAEs and diffusion models can be used to digest data distribution and get a new sample from the population. In Bond-Taylor et al. (2021) it was shown that GANs generation can achieve higher quality for high resolution images than VAEs. However, GANs can be unstable due to mode covering problem while VAEs are able to cover all modes of distribution.

Diffusion models Song et al. (2020) do not suffer from mode collapse and can beat GAN models in image quality. They are based on the idea of iterative refinement of an input noise signal until it converges to a specific data distribution, such as an image. Diffusion models are trained by the forward process of incremental noise injection into data image and sampling represents the inverse process of image generation from noise. However, they require a substantial time for training Xiao et al. (2021) as compared to GANs and VAEs.

To improve the inference time of diffusion models with minimal depreciation in quality flow matching for image generation, a method Lipman et al. (2022) was introduced that is based on the refinement of loss function of the continuous normalizing flows Mathieu & Nickel (2020). This model do not implicitly approximate the probability distribution but can produce high quality samples at reasonable time. Flow matching is more stable during training because of its loss function, making it a preferable choice for generation tasks compared to score-based diffusion methods, especially for low-dimensional data such as points of collocation for training PINNs. This is the reason why this model was used as a tool to add samples that move in the direction of large residual regions.

### 3.2 Adaptive sampling strategies for PINNs

In Tang et al. (2023a), the so called DAS PINN was proposed, in which a normalizing flow was applied for adaptive PDE residual sampling for solution of Poisson equation with singular source peaks, while in Wang et al. (2024a) a similar approach was used for a cavity flow problem. In Tang et al. (2023b) the Wassertein GAN-like model (WGAN) was proposed to solve the Poisson equation with narrow peaks in the source function. It generalizes a sampling approach for any normalized residual distribution $p_\alpha$ but also uses the KR-net architecture same, as in Tang et al. (2023a). The main difference of AAS-PINN Tang et al. (2023b) from DAS-PINN is that AAS PINN learns the distribution of residual with WGAN like a loss function that also applies regularization to the gradient of $p_\alpha$: $\nabla p_\alpha$, while Tang et al. (2023a) uses KL divergence loss.

Due to the fact that KR net model for probability density function (PDF) approximation is invertible, it implicitly models PDF. However let us note that it may face difficulties to approximate more complex probability density functions of points with large residual. Moreover, as it is articulated in Wang et al. (2024b) due to the fact that normalizing flows preserve the topology of the input space through continuous transformations, they face difficulties in representing certain simple classes of function Dupont et al. (2019).

This property leads to limited representation capabilities, high computational costs, and training problems in practical implementations Ho et al. (2019). That is why in this paper, we decided to use an architecture different from KR-net and other normalizing flow architectures that represent invertible transformations for our proposed method.

### 3.3 Normalizing Flow PINN

In Tang et al. (2023a), a model was developed, which integrates the PINN training with sampling from a normalizing flow. Subsequently, the flow model is refined through the minimization of the cross entropy loss between the residual and the flow's output logarithmic density. The flow model, implemented as KR-net, is constructed with affine coupling layers and the Knothe-Rosenblatt rearrangement. This architecture is uniquely designed to calculate both the forward and inverse probability density.

The optimization problem for the flow model is articulated as minimizing the Kullback-Leibler (KL) divergence between the residual function and the probability density generated by the flow. This

approach was specifically applied to solve the Poisson equation with a single peak source function and two peaks.

In the following section, we compare the normalizing flow PINN with our approach, providing a comprehensive analysis of the performance and efficiency pf sampling techniques.

### 3.4 FLOW MATCHING PINN

### 3.1 Flow matching

Flow matching Lipman et al. (2022) is the generative algorithm that can deal with high complexity of data. Unlike the normalizing flow method, it does not require the neural network transformation to be invertible. Instead of implicitly modeling the probability density function $p_1(x)$, the flow matching model enables sampling from this probability density function $p_1(x)$ by modeling vector field flow dynamics restoration. As sampling is based on sampling prior of Gaussian distribution $p_0(x)$ that though flow field dynamics $f_\theta(x, t)$ is transformed into more complex distribution $p_1(x)$, where time $t$ varies from 0 to 1. In this regard, the sample from unconditional probability density function can be generated with the solution of the ODE:

$$\begin{cases} dX_t = f(X_t, t)dt & , \\ X_0 \sim p_0 & . \end{cases} \qquad (4)$$

This fact is proved in Lipman et al. (2022) (see Theorem 1 therein) that relates the conditional probability distribution law dynamics $p_t(x|x_1)$ with vector field $f_\theta(x, t)$ through the continuity equation :

$$\frac{d}{dt} p_t(x) = -\operatorname{div}(p_t(x) f_t(x)) \qquad (5)$$

$$\int_0^1 E_{p_t(x)} ||f_\theta(x, t) - f(x, t)|| \qquad (6)$$

Let us note that $f(x, t)$ is unknown and this functional is not feasible to evaluate. It turns out that it can be reduced to the conditional flow dynamics $f_\theta(x, t|z)$, where $z$ can be a latent variable that is sampled from a prior distribution. That is why according to Theorem 2 Lipman et al. (2022) the intractable integral in 6 can be reduced to the following optimization problem that is tractable to solve:

$$\int_0^1 \mathbb{E}_{q(z)p_t(x|z)} ||f_\theta(x, t) - f(x, t|z)||^2 dt \to \min_\theta, \qquad (7)$$

The minimum that is a solution to this optimization problem is attained on the real vector field $f(x, t)$.

In order to find the minimum of such optimization problem the gradient can be calculated as an expectation that is found using the Monte Carlo approximation, namely,

$$\nabla_\theta \int_0^1 \mathbb{E}_{q(z)p_t(x|z)} ||f_\theta(x, t) - f(x, t|z)||^2 dt \qquad (8)$$

where $z \sim q(z), x \sim p_t(x|z)$. The equivalence of this fact was proved in Appendix that is based Theorem 2 from Lipman et al. (2022). Here we consider special case of optimal transport conditional vector field: $f(x, t|z) = (1 - (1 - \sigma_{min})t)x + tz$ for this we can rewrite the flow matching loss as:

$$\mathcal{L}_{CFM}(\theta) = \mathbb{E}_{t, q(x_1), p(x_0)} ||f_\theta(x, t) - (x_1 - (1 - \sigma_{\min})x_0)||^2$$

where $x_1$ belong to data sample. According Theorem 1 in Lipman et al. (2022) this marginal vector field produces the probability path that is justified through continuity equation 5.

### 3.4.1 THE SAMPLING ALGORITHM FOR PINN WITH BOOTSTRAP REWEIGHING

In order to tackle complex singularities in the domain (e.g. narrow peaks in the source function), we propose the following adaptive sampling algorithm for PINN. We apply flow matching paradigm as a vector field approximation of residual distribution approximation to sample points according to this vector field to add new points to the collocation point set for PINN training. The idea of the algorithm is based on refinement of PINN though adding points from region where high residual is concentrated. Instead of approximating the probability density function proportional to residual distribution we use this residual distribution in our algorithm to construct a probable sample from this distribution. For this case the residual of $r(x_i, \psi^k)$ is calculated from PINN:

$$r(x_i, \psi^k) = \mathcal{D}u(x_i, \psi^k) - s(x_i). \tag{9}$$

We construct a sub-sample $\mathbb{A}_i$ of size $M$ from previous sample $\mathbb{S}_{i-1}$ proportional to residual values using weighted bootstrap procedure. We train the vector field neural $f_\theta(x, t)$ net on this sub-sample $\mathbb{A}_i$ by minimizing flow matching objective 7 reformulated for tractability as in 8.

This vector field $f_\theta(x, t)$ governs the dynamics of point from prior distribution to the closest point in residual distribution. In this way by application of flow matching sampling we generate points according to this $f_\theta(x, t)$ by ODE dynamics to construct a new sample $\mathbb{V}_i$ :

$$\begin{cases} dX_t = f_\theta(X_t, t)dt &, \\ X_0 \sim p_0 & . \end{cases} \tag{10}$$

This ODE can be solved numerically by using the Euler-Maruyama discretization scheme.

Moreover, for efficient sampling of points we propose the following algorithm of solving PINN with generation point proposal based on flow matching form:

---
**Algorithm 1:** FMS PINN: PINN with matching flow

---
**Input** : number of points in initial sample $N$, number of points for training sample for vector field $M$, number of stages $K$

Sample $N$ points uniformly from the domain $\Omega$ denote this set as $\mathbb{S}_0 = \{\boldsymbol{x}_i\}_{i=1}^N$ ;
Train PINN model on sample $\mathbb{S}_0$ by optimizing empirical loss

$$\min_\psi L(\psi, N) = L_{PDE,N}(\psi) + L_{BC,N}(\psi) \tag{11}$$

where $L_{BC,N}(\psi)$ defined in 3 and $L_{PDE,N}(\psi)$ defined in 2. ;
**for** *k from 1 to K* **do**

    Calculate $r(\boldsymbol{x}_i, \_k)$ at each point of $\mathbb{A}_k$, i.e., and get values $\{r(\boldsymbol{x}_i, \psi^{k-1})\}_{i=1}^N$;
    Based on these weights, perform a weighted bootstrap resampling of points to form the root points $A_k$ for the flow, denoted as $\{\boldsymbol{x}_i^*\}_{i=1}^N$;
    Train the vector field $f_\theta(x, t)$ on this root points sample $A_k$ by optimizing flow matching objective as in 7;
    Sample new points $\mathbb{V}_k$ according to the vector field $f_\theta(x, t)$ that corresponds to $p_1(x)$ using the Euler method for solving ODE 10;
    Construct a sample for PINN as $\mathbb{S}_{k+1} = \mathbb{V}_k \cup \mathbb{S}_k$;
    Train PINN model $u(x, \psi^k)$ on $\mathbb{S}_{k+1}$ by optimizing loss 11 ;

**return** $u(x, \psi^k)$ ;

---

The sampling step of flow matching is performed following the steps listed below:

---
**Algorithm 2:** Flow-matching Sampling

---
**Input:** Trained network $f_\theta$, Sample-access to base distribution $q$, Step-size $\Delta t$
**Output:** Sample from target distribution $p$
$x_1 \leftarrow \text{Sample}(q)$;
**for** $t = 1, (1 - \Delta t), (1 - 2\Delta t), ..., \Delta t$ **do**
    $x_{t-\Delta t} \leftarrow x_t + f_\theta(x_t, t)\Delta t$;
**return** $x_0$;

---

The idea of Algorithm 1 can be represented by this diagram shown in Figure.

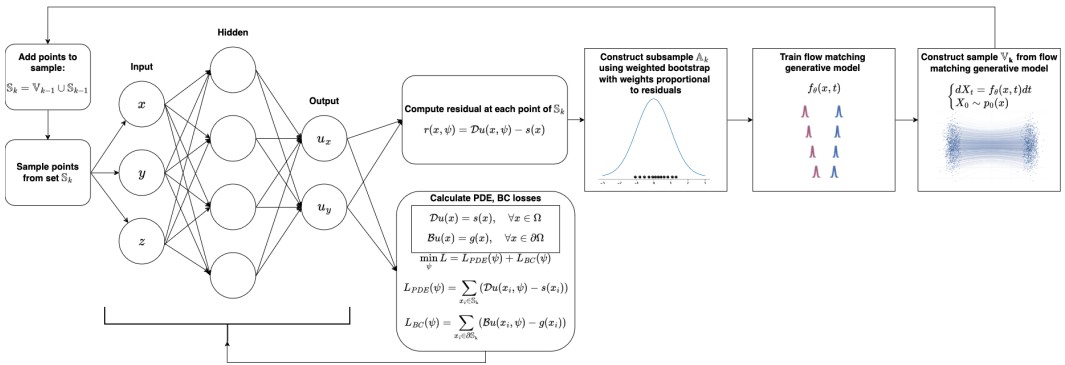

Figure 1: FMS PINN algorithm

# 4 NUMERICAL RESULTS

## 4.1 9 PEAKS PROBLEM

The issue of singular domains characterized by multi-modality in terms of peaks poses a significant challenge for simple generative models. These models, such as GANs, often struggle with the problem of mode collapse in terms of mode covering. This is why the Poisson equation with a source function consisting of 9 peaks is an important example to demonstrate the effectiveness of our method in enhancing PINN to address the problem of mode covering. By doing so, our method enables the production of accurate solutions that capture the complex structure of the domain.

The Poisson equation with 9 peaks looks as follows:

$$
\begin{aligned}
-\Delta u(\boldsymbol{x}) &= s(\boldsymbol{x}) \quad \text{in} \quad \mathbb{D}, \\
u(\boldsymbol{x}) &= g(\boldsymbol{x}) \quad \text{on} \ \partial\mathbb{D},
\end{aligned}
\tag{12}
$$

where $\boldsymbol{x} = [x_1, x_2]^\mathsf{T}$ and $\mathbb{D} = [-1,1]^2$. Here $s(\boldsymbol{x})$ has centers in $(x_0^i, y_0^i) = (-0.5, 0.5) + (\frac{mod(i,3)}{2}, 0) + (0, \frac{\lfloor i/3 \rfloor}{2})$, $i = 0, ..., 8$ and is represented by $s(\boldsymbol{x}) = \sum_{i=0}^{8} s_i(\boldsymbol{x})$, where

$$
\begin{aligned}
s_i(\boldsymbol{x}) &= -e^{-1000\left((x-c_{i,0})^2 + (y-c_{i,1})^2\right)} \left(\left(-2 \cdot 1000\,(x - c_{i,0})\right)^2 - 2 \cdot 1000\right) - \\
&\quad -e^{-1000\left((x-c_{i,0})^2 + (y-c_{i,1})^2\right)} \left(\left(-2 \cdot 1000\,(y - c_{i,1})\right)^2 - 2 \cdot 1000\right), i = 0, ..., 8
\end{aligned}
\tag{13}
$$

We also evaluate the absolute difference profile at the validation dataset for a network approximating the PDE solution. For the network, we used a fully connected network (FCN) with 6 blocks and a layer width of 64. For the flow matching model, we used an optimal transport coupling based on a FCN network. We trained the flow vector field model for 2000 iterations, each time resampling points and repeating the resampling every 5000 iterations, adding 28000 points each time. For

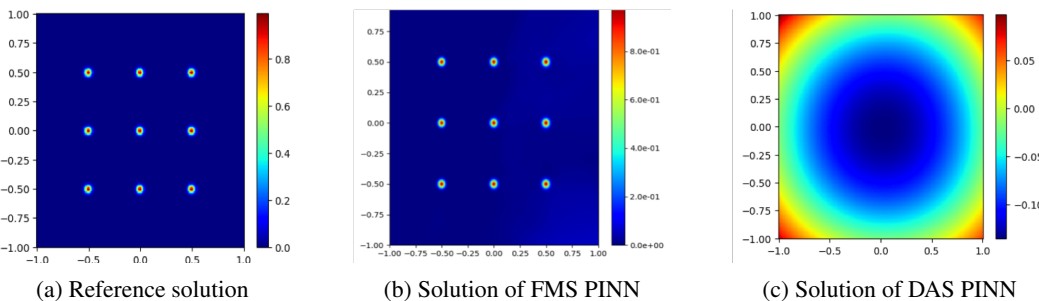

(a) Reference solution     (b) Solution of FMS PINN     (c) Solution of DAS PINN

Figure 2: Comparison of solution for 9 peaks problem

the normalizing flow comparison, we used the KR-net implementation from Tang et al. (2023a) in

TensorFlow with the same number of points, epochs, and resampling stages. We added the 9 peaks loss functional and reference in TensorFlow to make the comparison. In Figure 12, we observe that the normalizing flow method fails to capture the main solution compared to the matching flow approach. The flow matching PINN solution accurately depicts all nine peaks, keeping the solution outside the peaks close to zero. Figure 3 show that, in most of the domain, the solution of the flow matching PINN is quite accurate. The mean square difference (MSE) comparison between normalizing flow PINN and matching flow PINN during training for the 9 peak Poisson equation is illustrated by the MSE metrics calculated for different epochs and depicted in Figure 4b. The MSE of the Flow matching PINN decreases and converges to an order of $10^{-3}$. Our method samples points from peaks center as depicted by Figure 4a.

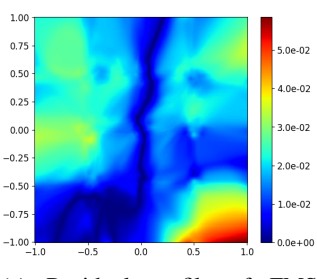 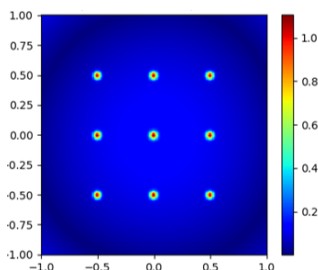

(a) Residual profile of FMS PINN

(b) Residual profile of DAS PINN

Figure 3: Comparison of residual profiles for 9 peaks problem

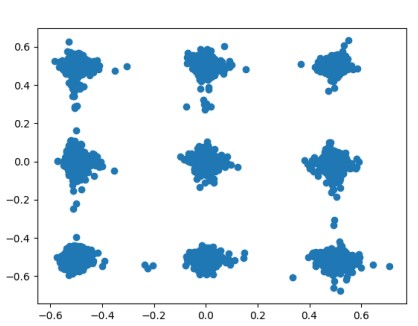 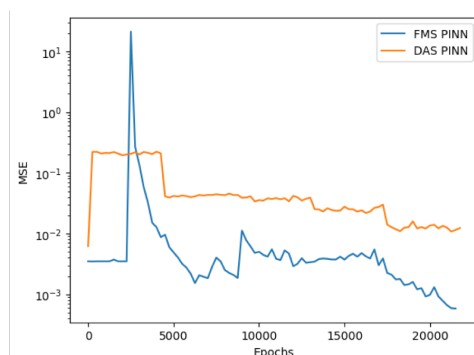

(a) Resampled points of FMS PINN added at 1 stage

(b) MSE comparison of FMS PINN with DAS PINN for 9 peaks problem

Figure 4: MSE plot and samples from FMS PINN

### 4.1.1 FIVE-DIMENSIONAL TWO PEAKS PROBLEM

For five dimensional problem the two centers of peaks are placed in $(x_1, x_2, x_3, x_4, x_5) = (0.5, 0.5, 0, 0, 0)$ and at $(x_1, x_2, x_3, x_4, x_5) = (-0.5, -0.5, 0, 0, 0)$. As in Zhang et al. (2024) the reference solution of this problem is:

$$u^*(x, y) = \sum_{i=1}^{c} \sum_{j=1}^{d} \exp\left[-K\left(\left(x_j - x_j^i\right)^2\right)\right], (x_1, x_2, \ldots, x_5) \in \Omega \tag{14}$$

where $K = 100$. In order to make an inference of the model and compute numerical errors efficiently we follow the methodology as in Zhang et al. (2024) where the two-stage sampling strategy for inference where proposed, where firstly 100k points are sampled uniformly across the domain. Then these points are combined with 15k points drawn from Gaussian distributions, whose mean and covariance are determined by each part of the solution led by one of the centers. These points

Table 1: Comparison of linear elasticity equation PINN with Normalizing flow PINN in terms of MSE

| Method | 2 peaks problem 16 | 2 peaks in 5D | 9 peaks problem |
|--------|--------------------|---------------|-----------------|
| FMS PINN | **7.7e-5** | **6.1e-3** | **4.2e-4** |
| DAS PINN | 5.2e-4 | 2.3 | 1e-1 |

with 10k points on the boundary are subsequently used to compute numerical errors for this five-dimensional two peaks problem.

For training purpose at initial step of training we draw 100k points from uniform distribution and 60k points from Gaussian centers. For optimizer algorithm we used Adam with learning rate 0.001.

We trained FMS PINN with sampling 40000 additional points at every resampling stage from the vector field trained via optimal transport flow matching objective on weighted bootstrap sub-sample from PINN training set $S_{k-1}$ and its residual distribution as weights.

We compare our algorithm with DAS PINN approach on the same number of training points equal to 100k and 60k points from center and for comparison use normalizing flow architecture of KR-net. We see that our method successfully captures all features of the solution, while method based on normalizing flow DAS PINN fails to produce the solution for same number of points and resampling stages.

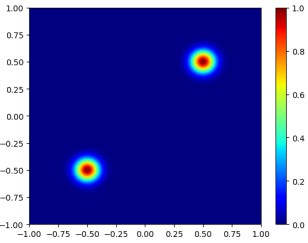 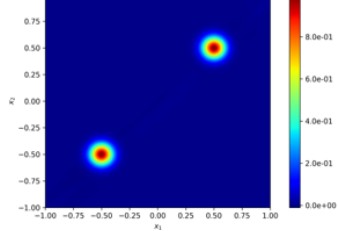 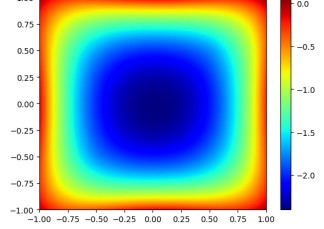

(a) Projection on first two coordinates of reference solution of 5D 2 peaks problem

(b) Projection of solution of 5D 2 peaks problem by FMS PINN algorithm

(c) Projection of solution of 5D 2 peaks problem by DAS PINN algorithm

Figure 5: 5D 2 peaks problem

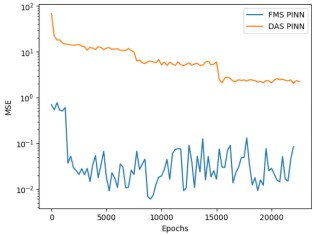 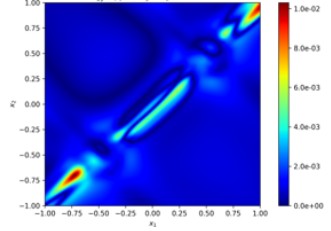 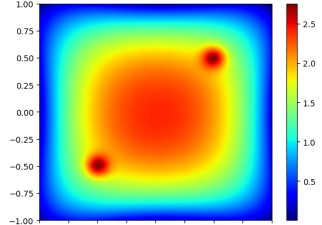

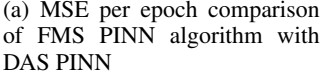 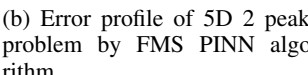

(a) MSE per epoch comparison of FMS PINN algorithm with DAS PINN

(b) Error profile of 5D 2 peaks problem by FMS PINN algorithm

(c) Error profile of 5D 2 peaks problem by DAS PINN algorithm

Figure 6: MSE comparison of FMS PINN algorithm with DAS PINN and comparison with reference

Finally, Table 1 summarizes comparison results for the normalizing flow PINN and the flow matching PINN for Poisson problems with peaks in source function, revealing comparable efficiency of the proposed method comparing to the normalizing flow.

## 4.2 LINEAR ELASTICITY EQUATION

In this section we consider solving a special instance of the mechanical equilibrium equation for a rectangular plate with a unique geometric inclusion made of a second material that we call linear elasticity equation. The primary equation that governs the mechanism of stress under deformation is

$$\nabla \cdot \sigma = 0,$$

where $\sigma$ is the stress tensor - the 2-nd order tensor describing the internal pressure state of the object. This equation is then can be represented:

$$
\begin{aligned}
C(1-\nu)\frac{\partial^2 u_x}{\partial x^2} + C\nu\frac{\partial^2 u_y}{\partial x \partial y} + \frac{1}{2}C(1-2\nu)\left(\frac{\partial^2 u_x}{\partial y^2} + \frac{\partial^2 u_y}{\partial x \partial y}\right) &= 0 \qquad \text{(x-axis)} \\
\frac{1}{2}C(1-2\nu)\left(\frac{\partial^2 u_x}{\partial x \partial y} + \frac{\partial^2 u_y}{\partial x^2}\right) + C\nu\frac{\partial^2 u_x}{\partial x \partial y} + C(1-\nu)\frac{\partial^2 u_y}{\partial y^2} &= 0 \qquad \text{(y-axis)},
\end{aligned}
\tag{15}
$$

where $E$ and $\nu$ are the Young modulus and Poisson ratio-constants, describing the material properties, while $u_x$ and $u_y$ represent horizontal and vertical displacement respectively.

The detailed derivation of this equation can be found in the subsection A.2.

where $C = \frac{E}{(1+2)(1-2v)} -$ constant.

We consider square plate with $(x, y) \in [x_{min}, x_{max}] \times [y_{min}, y_{max}]$. Dirichlet boundary conditions are enforced on horizontal displacement for the boundary of square.

$$
\begin{cases}
u_x(x, y) = -0.01, & x = x_{min}, \quad \forall y \in [y_{min}, y_{max}] \\
u_x(x, y) = 0.01, & x = x_{max}, \quad \forall y \in [y_{min}, y_{max}] \\
u_y(x, y) = 0.0, & \text{on the boundary}
\end{cases}
$$

We consider a specific kind of plate, that consists of one base material and second material in complex geometry inclusion, that is characterised with different Young modulus (= material property of stiffness) $E$. The geometric configurations are diamond and 2 circles.

Structure of the neural network is represented by 5 separate fully connected neural nets.

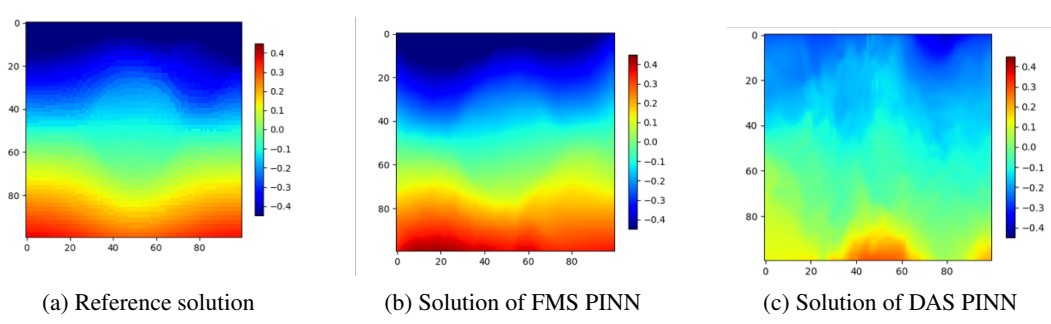

(a) Reference solution      (b) Solution of FMS PINN      (c) Solution of DAS PINN

Figure 7: Comparison of solution for 2 circles $u_x$ problem

For 30000 epochs of training we see that flow matching PINN outperforms DAS PINN. The results of MSE comparison for the flow matching PINN with the normalizing flow PINN is shown in Table 2. For two circles problem the flow matching method helps to improve quality, while for the diamond configuration it provides the solution of the same quality as the normalizing flow PINN.

Results of our method compared to the reference solution and DAS PINN for 2 circles case is illustrated in Figure 7 and Figure 8. PINN neural net architecture for our methods consists of 5 separate neural networks that have 5 fully connected layers with 40 neurons in each layer. As an optimizer, we use Adam with the scheduler ReduceLROnPlateau. As it is shown in Figure 9, our method captured all main patterns of the reference solution as wee see our algorithm FMS PINN outperforms DAS PINN.

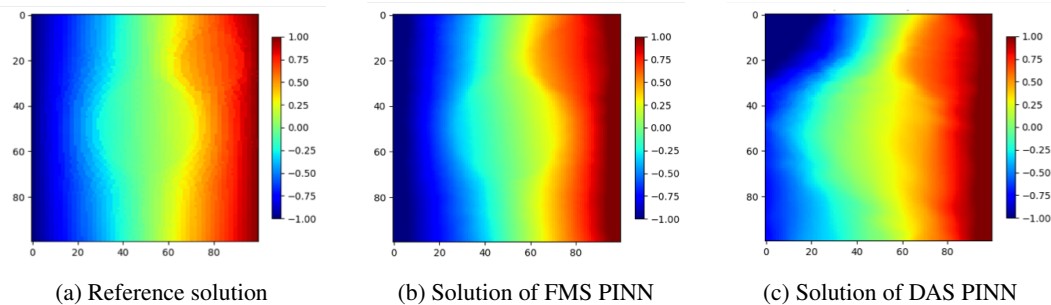

(a) Reference solution     (b) Solution of FMS PINN     (c) Solution of DAS PINN

Figure 8: Comparison of solution for 2 circles $u_y$ problem

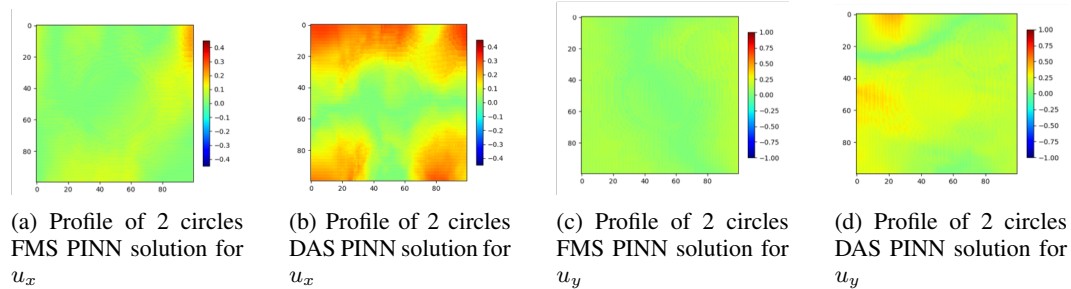

(a) Profile of 2 circles FMS PINN solution for $u_x$

(b) Profile of 2 circles DAS PINN solution for $u_x$

(c) Profile of 2 circles FMS PINN solution for $u_y$

(d) Profile of 2 circles DAS PINN solution for $u_y$

Figure 9: Comparison of error profiles for DAS PINN and FMS PINN for 2 cirlces problem

Table 2: Comparison of Elasticity PINN with Normalizing flow PINN in terms of MSE

| Method | 2 circles $u_x$ | 2 circles $u_y$ | diamond $u_x$ | diamonds $u_y$ |
|---|---|---|---|---|
| FMS PINN | **1.5e-3** | **7.9e-3** | **4.6e-3** | 9.2e-3 |
| DAS PINN | 1.7e-2 | 1.2e-2 | 7.1e-3 | **8.6e-3** |

## 5 CONCLUSION

In this paper a novel approach referred to as flow-matching sampling is proposed. It allows to select points for PINNs training, at which the evaluation of the PDE residual is performed. The idea of the method is based on the generative matching flows and adaptive sampling.

The numerical experiments show that our approach helps to solve singular problems and enhance the solution. We have examined an efficiency of the proposed method for the Poisson equation and linear elasticity equation system. It has been shown that the proposed method in several cases allow to achieve more accurate solution than the normalization flow approach. The latter can be considered as the closest competitor of the flow-matching method. It has been shown that the flow-matching method is efficient in the case of singularities in the solution. In our future work we will examine this method on larger number of epochs.

## 6 REPRODUCIBILITY STATEMENT

All of our experimental results are fully reproducible, and we have documented all settings and parameters used in our experiments. Upon request from the reviewers, we are prepared to provide the code and detailed instructions to help to replicate our findings. For the comparison with the DAS PINN method, we utilized the publicly available repository at https://github.com/MJfadeaway/DAS. By employing this repository, we ensured that our comparative analysis was conducted under consistent conditions, thereby guaranteeing a fair and accurate assessment between our proposed approach and the DAS PINN algorithm.

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

## A  APPENDIX

In this section, we apply the flow matching PINN and normalizing flow PINN to several PDEs. We begin our analysis with the Poisson equation that contains a singular source with peaks, representing the complex structure of the domain. As uniform sampling may not adequately cover the peak region within the domain, it can result in significant deviations of the PINN solution from the reference solution

### A.1  2 PEAKS PROBLEM

Next we consider an equation that has part that correspond to Laplace operator of the function that resembles Poisson equation combined with the divergence of the vector field of the function $u(\boldsymbol{x})\nabla v(\boldsymbol{x})$:

$$
\begin{cases}
-\nabla \cdot [u(\boldsymbol{x})\nabla v(\boldsymbol{x})] + \nabla^2 u(\boldsymbol{x}) = s(\boldsymbol{x}) & \text{in } \mathbb{D}, \\
u(\boldsymbol{x}) = g(\text{x}) & \text{on } \partial\mathbb{D}, \\
s(\boldsymbol{x}) = s_1(\boldsymbol{x}) + s_2(\boldsymbol{x}),
\end{cases}
\tag{16}
$$

$$
s_1(\boldsymbol{x}) = \left( e^{-1000(x_1-0.5)^2 + (x_2-0.5)^2} \right) \left( 4000(1000(x_1)^2 - 1000x_1 + 1000(x_2)^2 - 1000x_2 + 499) + \right.
$$
$$
\left. + 4(1000(x_1)^2 - 500x_1 + 1000(x_2)^2 - 500x_2 - 1)) \right.
$$

$$
s_2(\boldsymbol{x}) = e^{500\left(2x_1{}^2 + 2x_1 + 2x_2{}^2 + 2x_2 + 1\right)} \left( 4000(1000x_1{}^2 + 1000x_1 + 1000x_2{}^2 + 1000x_2 + 499) + \right.
$$
$$
\left. + 4(1000x_1{}^2 + 500x_1 + 1000x_2{}^2 + 500x_2 - 1)) \right.
$$

where $\boldsymbol{x} = [x_1, x_2]^{\mathsf{T}}$, $v(\boldsymbol{x}) = (x_1)^2 + (x_2)^2$, and the domain is $\mathbb{D} = [-1, 1]^2$. According to Tang et al. (2023a), the exact solution of 16 reads as follows:

$$
u(x_1, x_2) = e^{-1000[(x_1-0.5)^2 + (x_2-0.5)^2]} + e^{-1000[(x_1+0.5)^2 + (x_2+0.5)^2]},
\tag{17}
$$

which has two peaks at the points $(0.5, 0.5)$ and $(-0.5, -0.5)$. Here, the Dirichlet boundary condition on $\partial\Omega$ is given by the exact solution.

We see that our method succeeds in capturing the solution for the two-peaks problem and achieves the same order of error as compared to the normalizing flow PINN. Nevertheless, the MSE score is 1.5 times lower for the matching flow PINN as compared to the normalizing flow PINN.

To improve the flow matching model, we trained it for 2,000 iterations each time we resampled points and repeated the resampling process every 10,000 iterations, adding 28,000 points during each resampling stage. In the initial stage to prevent overfitting, we trained the initial PINN model for 10,000 epochs. For comparison of our flow matching with the normalizing flow we use a KR-net implementation from Tang et al. (2023a) code implementation in Tensorflow with same number of points, epochs and re-sampling stages. Figure 10 indicates that after 6000 epochs the matching flow algorithm achieves better quality than normalizing flow PINN. For this problems flow matching PINN after 25000 is better than solution of the normalizing flow PINN, while Figure 11 shows that both normalizing flow PINN and flow matching PINN approximate the solution in a descend way.

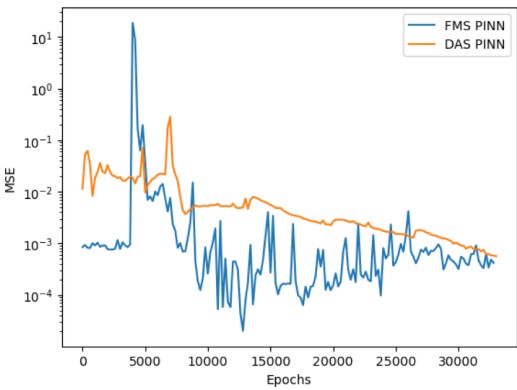

Figure 10: MSE comparison for 2 peaks problem

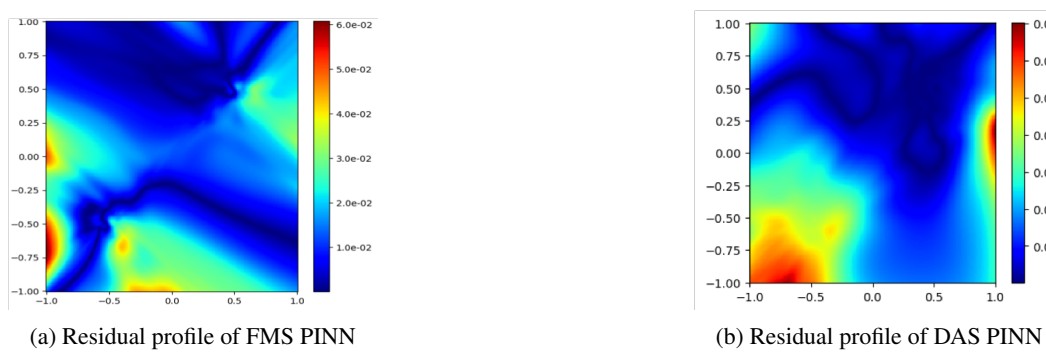

(a) Residual profile of FMS PINN

(b) Residual profile of DAS PINN

Figure 11: Comparison of residual profiles for 2 peaks problem

## A.2 LINEAR ELASTICITY EQUATION: DETAILS

Here is more detailed derivation of linear elasticity equation. The primary equation that governs the mechanism of stress under deformation is

$$\nabla \cdot \sigma = 0,$$

where $\sigma$ is the stress tensor - the 2-nd order tensor describing the internal pressure state of the object. In matrix notation, equation for the 2D case takes the form:

$$\begin{pmatrix} \frac{\partial}{\partial x} \\ \frac{\partial}{\partial y} \end{pmatrix} \cdot \begin{pmatrix} \sigma_{xx} & \sigma_{xy} \\ \sigma_{xy} & \sigma_{yy} \end{pmatrix} = 0, \tag{18}$$

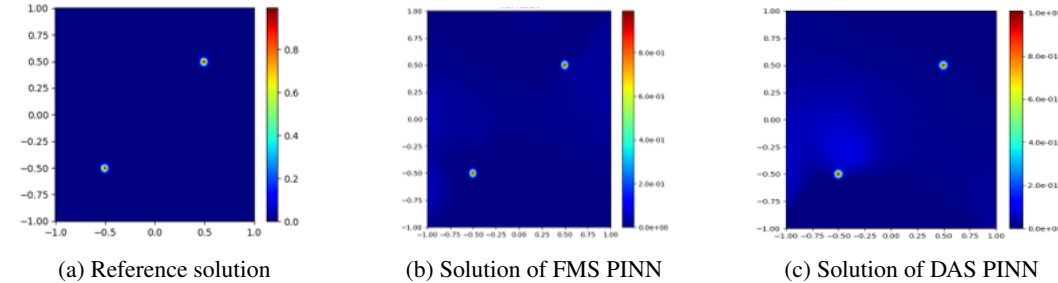

(a) Reference solution  (b) Solution of FMS PINN  (c) Solution of DAS PINN

Figure 12: Comparison of solution for 2 peaks problem

We can express the stress tensor $\sigma$ in terms of the $4^{th}$ order elasticity tensor $C$ and th $2^{nd}$ order strain tensor $\epsilon$ :

$$\sigma_{ij} = C_{ijkl} \cdot \epsilon_{lk}$$

For the case of linear isotropic materials in 2D, this reduces to 2-dimensional Hooke's law for plane strain:

$$\begin{pmatrix} \sigma_{xx} \\ \sigma_{yy} \\ \sigma_{xy} \end{pmatrix} = \frac{E}{(1+\nu)\cdot(1-2\nu)} \begin{pmatrix} 1-\nu & \nu & 0 \\ \nu & 1-\nu & 0 \\ 0 & 0 & 1-2\nu \end{pmatrix} \cdot \begin{pmatrix} \epsilon_{xx} \\ \epsilon_{yy} \\ \epsilon_{xy} \end{pmatrix}, \tag{19}$$

where $E$ and $\nu$ are the Young modulus and Poisson ratio constants, describing the material properties, $E$ and $\nu$ are individual for each material and can vary a lot from one material to another

The strain tensor $C$ describes the deformation of the solid body in each elementary volume. The $\epsilon_{xx}$ and $\epsilon_{yy}$ components are describing the relative elongation of the elementary volume in $x$ and $y$ directions respectively, and the $\epsilon_{xy}$, component describes the shift deformations.

In classical case the equation is solved with respect to displacements, and the strain tensor can be expressed:

$$\epsilon_{xx} = \frac{\partial u_x}{\partial x},$$
$$\epsilon_{yy} = \frac{\partial u_y}{\partial y}, \tag{20}$$
$$\epsilon_{xy} = 0.5 \left( \frac{\partial u_x}{\partial y} + \frac{\partial u_y}{\partial x} \right).$$

where $u_x$ and $u_y$ are the displacements of the points of solid body in directions $x$ and $y$ respectively

By substituting this idea 18 to initial equation 20 we can finally obtain:

$$C(1-\nu)\frac{\partial^2 u_x}{\partial x^2} + C\nu\frac{\partial^2 u_y}{\partial x \partial y} + \frac{1}{2}C(1-2\nu)\left(\frac{\partial^2 u_x}{\partial y^2} + \frac{\partial^2 u_y}{\partial x \partial y}\right) = 0 \quad \text{(x-axis)}$$
$$\frac{1}{2}C(1-2\nu)\left(\frac{\partial^2 u_x}{\partial x \partial y} + \frac{\partial^2 u_y}{\partial x^2}\right) + C\nu\frac{\partial^2 u_x}{\partial x \partial y} + C(1-\nu)\frac{\partial^2 u_y}{\partial y^2} = 0 \quad \text{(y-axis)} , \tag{21}$$

where $C = \frac{E}{(1+2)(1-2v)}$ − constant

We consider square plate with $(x, y) \in [x_{min}, x_{max}] \times [y_{min}, y_{max}]$. Dirichlet boundary conditions are enforced on horizontal displacement for the boundary of square.

$$\begin{cases} u_x(x,y) = -0.01, & x = x_{min}, \quad \forall y \in [y_{min}, y_{max}] \\ u_x(x,y) = 0.01, & x = x_{max}, \quad \forall y \in [y_{min}, y_{max}] \\ u_y(x,y) = 0.0, & \text{on the boundary} \end{cases}$$

We consider a specific kind of plate,that consists of one base material and second material in complex geometry inclusion, that is characterised with different Young modulus $E$. The geometric configurations are diamond and 2 circles.

Here is the solution for diamond configuration for FMS PINN compared with DAS PINN:

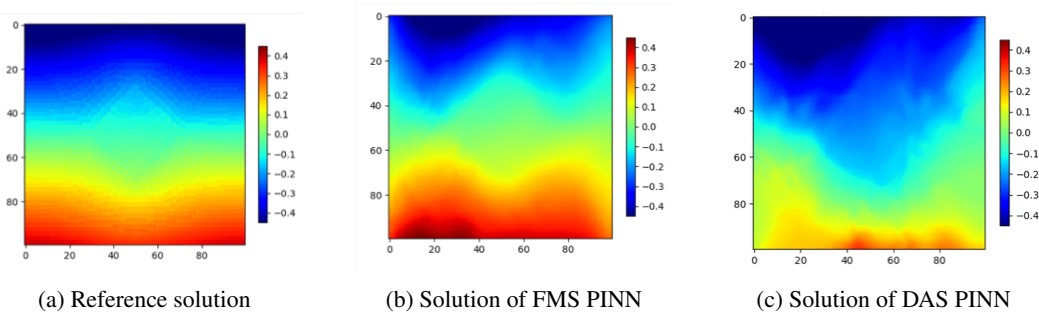

(a) Reference solution  (b) Solution of FMS PINN  (c) Solution of DAS PINN

Figure 13: Comparison of solution for diamond $u_x$ problem

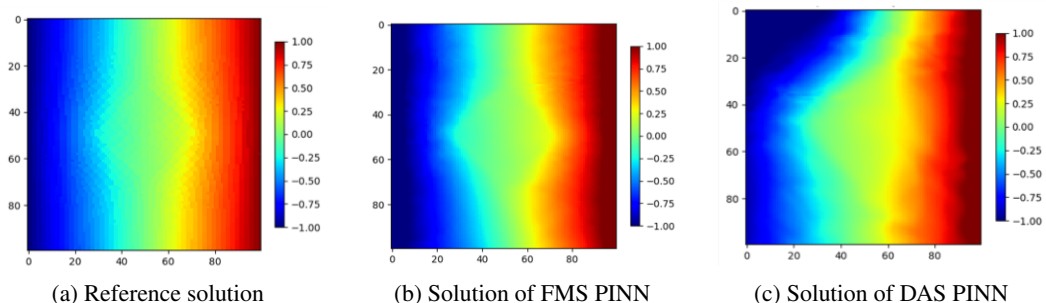

(a) Reference solution  (b) Solution of FMS PINN  (c) Solution of DAS PINN

Figure 14: Comparison of solution for diamond $u_y$ problem

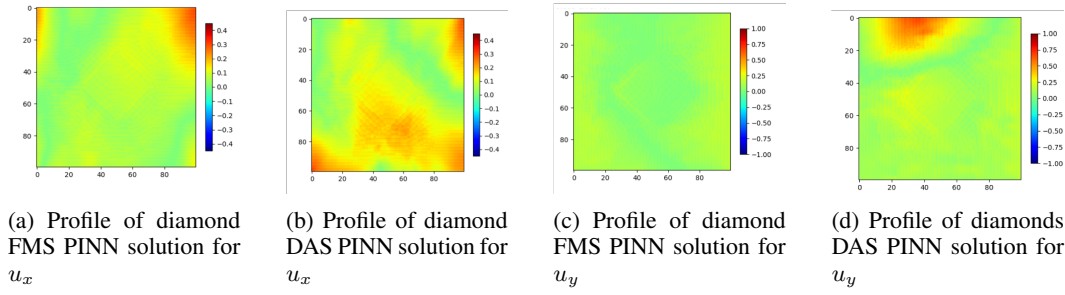

(a) Profile of diamond FMS PINN solution for $u_x$

(b) Profile of diamond DAS PINN solution for $u_x$

(c) Profile of diamond FMS PINN solution for $u_y$

(d) Profile of diamonds DAS PINN solution for $u_y$

Figure 15: Comparison of error profiles for DAS PINN and FMS PINN for diamond problem

Here is comparison in terms of MAE for 2 configurations: 2 circles and diamond.

We also see that our method outperforms DAS PINN in terms of sum MSE for $u_x$ and $u_y$ for both diamond and 2 circles setups as depicted by Figure 16:

Table 3: Comparison of Elasticity PINN with Normalizing flow PINN in terms of MAE

| Method | 2 circles $u_x$ | 2 circles $u_y$ | diamond $u_x$ | diamonds $u_y$ |
|---|---|---|---|---|
| FMS PINN | 0.04184 | 0.07817 | 0.073 | 0.097 |
| DAS PINN | 0.12 | 0.17 | 0.10 | 0.14 |

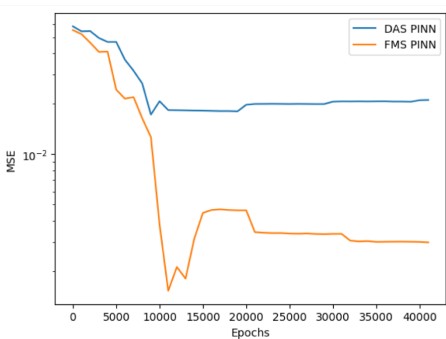

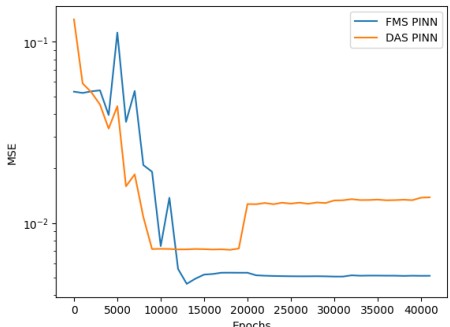

(a) MSE per epoch for 2 circles setup of linear elasticity equation

(b) MSE per epoch for diamond setup of linear elasticity equation

Figure 16: Comparison of MSE for diamond and 2 circles setup for FMS PINN and DAS PINN

