# OpenReview forum: "FMS PINN: Flow-matching sampling for efficient solution of partial differential equations with source singularities"
_ICLR.cc/2025/Conference — Submitted to ICLR 2025_

### Official Review · Reviewer_g3T5 · 2024-10-15

**Soundness:** 1
**Presentation:** 1
**Contribution:** 1
**Rating:** 3
**Confidence:** 3

**Summary:**

This paper proposes a method using flow matching for adaptive sampling of residual points in physics-informed neural networks (PINNs), aimed at improving stability and accuracy, particularly for PDEs with singularities. The approach is compared against normalizing flow-based sampling, demonstrating advantages in certain scenarios.

**Strengths:**

The paper demonstrates improvements of the flow matching-based sampling method over the previous normalizing flow approach. The proposed methodology might provide insights for the community.

**Weaknesses:**

- The writing could be improved. Many citations should use `\citep` instead of `\citet`. Some symbols are not properly defined, e.g. $\mathbb{S_k}$ appears in Equations (2) and (3), long before it is introduced in Algorithm 1. The definition of the optimization problem in Equation (7) is nonstandard. Additionally, some acronyms are not properly introduced, such as DAS and AAS.
-  The structure of the paper is confusing. Section 3.4 appears to describe the author's original work but falls under "3. Related Work." Furthermore, there is a "Section 3.1" comes after Section 3.4, which disrupts the logical flow.
- Some figures have random gray lines surrounding them (e.g., Figures 2(b,c) and 4(b), among others).
- The novelty of the proposed method seems limited. The work replaces GAN or normalizing flow in previous approaches with flow matching, which is known to have some advantages in certain scenarios. This incremental improvement may not be sufficient to justify the claims of significant advancement.
- While the work might be useful, its practical impact on solving PDEs appears limited. Practitioners typically care about whether they should use this method for solving their PDE problem, including aspects like accuracy and computational speed. The paper primarily focuses on showing that this ML method is more accurate than a previous ML method, without adequately addressing the broader context of whether PINNs are appropriate for the given problem compared to traditional numerical methods.
- The use of the term "source singularities" might be inappropriate. The source terms in the Poisson equations presented are sharp Gaussians at best, not truely singularities (delta function). These singularities can lead to discontinuous gradient or solutions. See methods for elliptic interface problems. Even in PINN, there are more exact method to handle the singularities, e.g, Tseng, Y.-H., Lin, T.-S., Hu, W.-F., Lai, M.-C., 2023. A cusp-capturing PINN for elliptic interface problems. Journal of Computational Physics.
- The two inset figures in Figure 1 ("train flow matching generative model" and "construct sample from flow matching generative model") do not appear to be original work. Similar images can be found in Figures 7 and 21 on [this website](https://mlg.eng.cam.ac.uk/blog/2024/01/20/flow-matching.html).
- The authors should provide a discussion on computational time. For instance, in Section 4.1, 28,000 points are introduced at each resampling stage, leading to a substantial increase in point count during the whole training process. Since"high computational costs" was mentioned as a drawback of previous approaches, it is unclear that the proposed method can alleviate this issue.
- The authors should report the mean and standard deviation of different trials, as sampling is required in this method. This would provide a better understanding of the variability and reliability of the results.

**Questions:**

- In Algorithm 2, why is the ODE solved backward from $t = 1$ to $t = 0$? This seems to contradict Equation (10). Could the authors clarify this?
- Based on Figure 4(a), it is unclear why a similar distribution could not be achieved using simpler methods like RAR (or other methods that do not require a neural network) or the weighted bootstrap step in this work. What happens if we remove the flow matching step?
- The authors mention both AAS and DAS but only compare their method with DAS. Since AAS claims to improve upon DAS, how does the proposed method compare to AAS?
- In Table 1, the accuracy of DAS is reported to be on the order of 1, which seems lower than values cited in the literature (around 1e-2, see Table 1 in AAS paper and table 1 and 2 in DAS paper). Could the authors explain this discrepancy?
- In Section 4.2, it would be helpful if the authors showed the material interface and the final distribution of collocation points, allowing readers to better understand how the proposed method works.

---

### Official Review · Reviewer_W7jW · 2024-10-29

**Soundness:** 2
**Presentation:** 2
**Contribution:** 2
**Rating:** 3
**Confidence:** 2

**Summary:**

This paper proposes a resampling procedure for PINN training when the right-hand side is sharply peaked.  The idea is to use flow matching to generate new samples in regions where the residual is high; in each iteration, more quadrature points are added to the PINN objective function to penalize points where the PDE is harder to solve.  The method is tested on the Poisson equation and a simple version of elasticity.

**Strengths:**

Resampling for PINNs makes sense and can help identify regions where the solver needs extra "help" to find the right solution.

**Weaknesses:**

Overall, the method seems to be a fairly heuristic extension of existing works.  Although I see the authors are applying ideas from the flow matching work, they do not articulate important details of their algorithm, e.g. the distribution from which flow matching is attempting to draw samples.  The tests here show preliminary evidence that the method has some value but are far from conclusive (just 3 PDE examples total as far as I can tell).

**Questions:**

I have also included comments in this section.

***

Careful with LaTeX formatting bugs --- e.g., in first paragraph of the introduction there is a stray period (line 38), backward quotation marks (line 42), and \citet{} style citations that should be \citep{}.  There are also some mild grammar issues (e.g., missing “the” before “PINN loss function” on line 43).  These didn’t impede understanding, but I would suggest a thorough edit before the final version of this paper is published.

In the unnumbered equation above (2), should the norms be squared?  And do you need a parameter trading off between the interior and boundary terms?

Should the sums in (2)-(3) be integrals?  Are the samples x_i re-drawn in every iteration or kept fixed during the training procedure?

The last three paragraphs of section 3.1 seem like they’re missing a connection to what came before them.  In particular, these paragraphs describe generic methods for variational inference and sampling, but it’s not entirely clear how these methods are applied to PINNs specifically.

Line 166:  “pf”

Line 171, “the generative algorithm” --- one of many

Should the loss in (6) be squared?

It seems a lot of section 3.1 is repeating what’s in the flow matching paper.  Is all of this discussion needed to describe the proposed new algorithm in the paper?

Exposition-wise, I would suggest reallocating the “real estate” in the paper’s discussion substantially.  In particular, the first 4 pages of the paper are background, then there is exactly 1 page describing the method, and then the remainder of the paper describes results.  It seems valuable to extend the discussion of the proposed method, making sure the algorithm is described in full detail and that appropriate properties of the sampling method (e.g., making sure it recovers solutions to the PDE as the neural network gets more neurons) are stated carefully.  The exposition of the algorithm in section 3.4.1 is quite terse --- for example, precisely what distribution is the flow matching method sampling from?

Eq (10) is deterministic other than the initial condition X_0 --- there is no Brownian motion term, for example.  Why use stochastic PDE notation?

Algorithm 2 seems to be standard “forward Euler” solution of an ODE and can probably be omitted.

Algorithm 1 has many vague steps.  For example, what algorithm is used for weighted bootstrap resampling?  How is the vector field “trained” and for how many steps?  Do you prune the set of points S_k in each step or does it get larger and larger in each iteration?

Figure 1 did not help me understand the algorithm and seems quite abstract.

Is it possible to replace the peaks in example (13) with true delta functions, or does the right-hand side of the Poisson equation in the proposed method need to be differentiable?

---

### Official Review · Reviewer_SCKW · 2024-10-31

**Soundness:** 2
**Presentation:** 2
**Contribution:** 2
**Rating:** 3
**Confidence:** 2

**Summary:**

The authors propose a new strategy for re-sampling new training data points for training PINN networks. At each stage of the PINN training algorithm, they train a new flow-based generative model for sampling points in regions where the PDE residuals have large values. At each epoch, the training points are the union of the previous points and the new generated points.

**Strengths:**

- Both the objective and the proposed algorithm are well explained.
- Most of the related works and useful concepts are explained.
- The contribution makes sense. Indeed, using normalizing flows for resampling points seems to be an overkill, as it is not necessary to have explicit probability densities and invertibility.
- Different experimental conditions

**Weaknesses:**

- You only compare with DAS PINN and no other methods. As you mention AAS-PINN and RAR in the related works, you should also compare with these methods. Especially, it would be interesting to have comparisons with methods that do not use generative models for re-sampling.
- Moreover I am questioning how fair is your comparison with DAS PINN. Indeed, you use the same number of training epochs / training points for both models, but both do not share the same architecture. It seems in Figure 4 b) and 10 that DAS-PINN has not fully converged yet. Can you provide the same curves with a larger number of epochs for both methods ?
- The "weighted bootstrap resampling " algorithm should be clearly explained and detailed. It is a major block of the method, and it is mentioned without further explanations.
- There are several typos, missing commas, english mistakes that need to be corrected.
- Color inconsistency in the plots of Figure 16.
- Sections 3.2 and 3.3 seem to repeat / should be merged.
- $p_1(x)$ is used without being defined. Is it a pushforward measure ?
- The  explanation of the AAS method is not clear enough for me.

**Questions:**

- Is $f_\theta$ re-trained from scratch at each iteration, or fined-tuned ?
- It seems that the number of training points grows with the number of iterations. Is it really the case, or do you discard training points at each iteration ?
- How does the method scale to higher dimension ? It seems that the advantage of using flow-based generation over normalizing flow should be stronger in high dimension. Unfortunately, the authors experiment only up to dimension 5.

---

### Official Review · Reviewer_8sXG · 2024-11-05

**Soundness:** 2
**Presentation:** 2
**Contribution:** 2
**Rating:** 3
**Confidence:** 4

**Summary:**

This paper presents a method for automatic and adaptive sampling of domain points during the training of a physics-informed neural network (PINN) model. The main idea is to use a flow-matching technique to learn the distribution of the ''correct'' density of the domain points from the current residuals. This is similar to the ''mesh refinement'' procedures common in PDE applications. At a typical ''refinement'' step in the current method, the residuals are used to train a flow-matching model to arrive at the unknown distribution of the domain points. Now, this model is used to generate point samples for the next loop of training. The authors present two examples with singular behaviour, and  compare their results with one other work, which is called ''DAS-PINN.'' In both these cases, "FMS-PINN" is seen be the superior method.

`Overall impression:` The major idea, i.e., using a generative model to generate PINN points for training is not new. The marginal contribution here is using flow-matching technique to do so. Nevertheless, it makes for an interesting read, and the results, though neither comprehensive nor conclusive, are interesting as well. Decent paper, but not ready for publication.

**Strengths:**

* The paper is mostly well written, but some parts are confusing (see questions).
* Interesting application of flow-matching to generate sample points for PINN training.

**Weaknesses:**

* The mathematical exposition (Sec 3.4) is not well written (see questions).

* The authors compare their results with one existing method (DAS-PINN, Tang et al., 2023). But only one of the examples from Tang et al. is recreated in this paper, and that too is relegated to the appendix. This is the only point of exact comparison, and it is inconclusive whether one method is better than the other.

* (minor) Line 325: (probably) incorrectly refers to Figure 12

**Questions:**

* Please rewrite the first subsubsection under Sec 3.4 (titled ''3.1 Flow matching'' for some reason!). The probability densities $ p_0 $ and $ p_1 $ are used without any introduction. Line 175: :''As sampling is based on ..." is unintelligible. Equation 6 is written without any preceding or succeeding text. Equation 7 can be written in much simpler notation (a standard minimization statement can be used). The whole subsection feels very disjointed. I think, rather than attempting a generic text on flow-matching, this subsection can be used for a better exposition to the particular application you are aiming for.

* Sec 3.4.1 presents the main contribution of this paper, but it feels hurried. Please explain the ''weighted bootstrap procedure'' in detail, or cite appropriate references.

* Sec 4.1: How do the results compare if you simply sample more points near the peak, since the peaks are known a priori?

* Sec 4.2: please provide a better description of the problem, e.g., with a schematic diagram, or cite a reference text.

* All the presented examples are where the ''singularity'' regions are known a priori. How would this method work when a PDE solution naturally develops regions that require refinement, e.g., flow problems that develop boundary layers. This can be shown by taking a simple advection-diffusion equation with Dirichlet boundary conditions.

* How was the DAS-PINN code configured? What were the hyper-parameters? As someone who has not run either of the codes, how can I convince myself that due diligence was done on the DAS-PINN hyperparameters?

---

### Meta-Review · Area_Chair_dUjE · 2024-12-20

**Metareview:**

Reviewers agree to reject this paper with its current state due to lack of novelty, inconsistencies in text, and missing clarifications.
Authors did not respond to the reviewer comments. Therefore the paper is recommended for rejection.

**Additional Comments On Reviewer Discussion:**

Authors did not rebut.

---

### Decision · Program_Chairs · 2025-01-22

Reject